# UNCERTAINTY-AWARE OPTIMIZATION VIA ONLINE BOOTSTRAPPING

## ABSTRACT

Standard gradient descent methods yield point estimates with no measure of confidence. This limitation is acute in overparameterized and low-data regimes, where models have many parameters relative to available data and can easily overfit. Bootstrapping is a classical statistical framework for uncertainty estimation based on resampling, but naively applying it to deep learning is impractical: it requires training many replicas, produces post-hoc estimates that cannot guide learning, and implicitly assumes comparable optima across runs—an assumption that fails in non-convex landscapes. We introduce Twin-Bootstrap Gradient Descent, a resampling-based training procedure that integrates uncertainty estimation into optimization. Two identical models are trained in parallel on independent bootstrap samples, and a periodic mean-reset keeps both trajectories in the same basin so that their divergence reflects local (within-basin) uncertainty. During training, we use this estimate to sample weights in an adaptive, data-driven way, providing regularization that favors flatter solutions. In deep neural networks and complex high-dimensional inverse problems, the approach improves calibration and generalization and yields interpretable uncertainty maps.

## 1 INTRODUCTION

Modern machine learning faces a tension between efficient optimization and uncertainty quantification. Gradient-based training drives parameters to a single point estimate but provides little information about confidence in that estimate—especially for large, overparameterized models trained with limited data, where overfitting and miscalibration are common. Many approaches handle uncertainty post hoc, so the training process itself is not guided by uncertainty estimates. The need is for an online, model- and data-aware signal that can inform learning while it happens, at the scale of modern deep networks.

Statistical resampling methods such as bootstrapping are typically used after training, separate from optimization, so information from finite-sample variability does not shape learning. We instead integrate a resampling-based uncertainty estimate directly into the optimization loop. Classical bootstrapping provides a robust estimate of parameter uncertainty but is impractical at scale because it requires retraining many replicas; it also yields only post-training estimates. An online estimate during training can act as a regularizer, guiding the search toward more generalizable solutions.

The challenge of adapting bootstrapping to modern deep learning is not just computational; it is also conceptual. The non-convex, multi-modal loss landscapes of neural networks mean that independently trained models can converge to entirely different, yet equally valid, solutions. In such a scenario, the divergence between model parameters is a meaningless measure of uncertainty, reflecting inter-basin distance rather than local landscape geometry.

We present a training procedure that integrates a resampling-based uncertainty estimate directly into gradient descent. The approach has three components: (1) using independently bootstrapped datasets during optimization to obtain an online uncertainty signal that guides learning; (2) a two-model design that limits cost while retaining the benefits of resampling; and (3) a periodic mean-reset that keeps both models in the same solution basin so their divergence measures local (not inter-basin) uncertainty.

The uncertainty estimate is derived from the variability present in finite datasets via a principled resampling mechanism and requires no modification to the task loss. We recast classical bootstrapping as an *online two-sample estimator*, turning a post-hoc analysis into a signal that directly informs optimization.

## 2 RELATED WORK

This work connects optimization with uncertainty quantification by combining gradient-based training and a resampling-based uncertainty signal. We contrast it with existing work along three axes: the purpose of uncertainty, the underlying theoretical framework, and the method of optimization.

### 2.1 POST-HOC UNCERTAINTY VS. ONLINE REGULARIZATION

Many popular methods treat uncertainty quantification as a post-hoc analysis, which is fundamentally limited as it cannot influence the training process itself. Deep Ensembles are a prime example, where multiple independent models are trained from scratch and their predictions are averaged to estimate predictive uncertainty (Lakshminarayanan et al., 2017). While effective at capturing multimodal solutions, this approach is computationally expensive and provides no mechanism for per-step regularization. Similarly, classical bootstrapping, while statistically robust, requires numerous independent model trainings, making it impractical for modern large-scale networks. In contrast, an online, parameter-level uncertainty estimate can be used to regularize the model at every step, directly linking uncertainty to optimization.

### 2.2 BAYESIAN VS. DATA-DRIVEN UNCERTAINTY

The canonical approach to uncertainty is Bayesian, where the goal is to learn a posterior distribution over model weights. Bayesian Neural Networks (BNNs) (Blundell et al., 2015; Wilson & Izmailov, 2020) and their approximations, like Monte Carlo Dropout (Gal & Ghahramani, 2016), are powerful but rely on strong assumptions about prior distributions and the form of the approximate posterior. Furthermore, methods that view Stochastic Gradient Descent and its variants as approximate samplers are elegant in their theoretical framing (Welling & Teh, 2011; Chen et al., 2014; Mandt et al., 2017). They argue that SGD's mini-batch noise acts as a form of thermal noise, helping the optimizer explore the parameter space and sample from the posterior. However, for this to be a principled approach, the learning rate must be precisely tuned to the local curvature of the loss landscape, as defined by the Hessian matrix, to ensure correct sampling. Computing the Hessian is computationally prohibitive for large-scale models, making this approach difficult to apply in practice. In contrast, the present work uses a resampling-based uncertainty signal derived from finite datasets during training, requires no modification to the task loss, and introduces explicit, controlled stochasticity. As a result, it does not rely on Hessian information or incidental optimizer noise and can be paired with standard gradient-based optimizers, including full-batch training.

### 2.3 GRADIENT-BASED VS. DERIVATIVE-FREE OPTIMIZATION

Uncertainty can be used to guide optimization. There is a conceptual link to algorithms like Covariance Matrix Adaptation Evolution Strategy (CMA-ES) (Hansen & Ostermeier, 2001; Hansen, 2016), which adapt exploration using covariance information. However, such derivative-free methods are significantly slower on high-dimensional problems. Here, uncertainty is estimated during gradient-based training, providing a landscape-aware signal without sacrificing efficiency.

## 3 THE TWIN-BOOT METHOD

### 3.1 PROBLEM SETUP AND NOTATION

Let the dataset be $D = \{(x_i, y_i)\}_{i=1}^N$ drawn i.i.d. from an unknown distribution $P_{\text{data}}$. We consider models $f(x; w)$ with parameters $w \in \mathbb{R}^P$, trained to minimize an empirical loss

$$L(w; D') = \frac{1}{|D'|} \sum_{(x,y) \in D'} \ell\big(f(x; w), y\big), \quad D' \subseteq D.$$

Parameters may be partitioned into groups indexed by $\ell$ (e.g., layers) with sizes $D_\ell$; we maintain group-wise uncertainty estimates $\sigma_\ell^2$ during training. Two bootstrap datasets $D_1^*$ and $D_2^*$ are formed by sampling with replacement from $D$.

## 3.2 A Primer on Classic Bootstrapping for Uncertainty

We briefly recall the principles of classical bootstrapping that motivate our approach. Given a true, but unknown, underlying data distribution $P_{\text{data}}$, the true parameter uncertainty is defined as the variance of the optimal parameters obtained across all possible finite datasets drawn from this distribution (Efron, 1979; Efron & Tibshirani, 1994). Mathematically, it is the variance of the empirical risk minimizer:

$$\text{Var}_{D \sim P_{\text{data}}} \left( \arg\min_w \frac{1}{|D|} \sum_{(x,y) \in D} L(w; x, y) \right).$$

Since this quantity is impossible to compute, classical bootstrapping provides a robust, non-parametric method for estimating it (Efron & Tibshirani, 1994). The process involves creating a large number of resampled datasets, known as bootstrap samples, by drawing $N$ data points with replacement from the original dataset $D$. For each bootstrap sample $D_b^*$, a new model is trained from scratch, yielding a parameter vector $w_b^*$. The collection $\{w_1^*, \ldots, w_B^*\}$ forms a sampling distribution whose variance, $\text{Var}(w^*)$, is a statistically robust estimate of the true parameter uncertainty.

**Practical considerations and design choices.** This presents three obstacles for modern deep learning, and we address each with a targeted design choice:

- **Computational cost.** Classical bootstrapping requires training many replicas. *Solution:* a *two-sample estimator* with only two twins, yielding about a $2\times$ overhead.
- **Non-convexity and multi-modality.** Independent replicas drift to different minima, so variance reflects inter-basin distance rather than local uncertainty. *Solution:* a periodic *mean-reset* that confines twins to the same basin so their divergence measures local uncertainty.
- **Post-hoc nature.** Classical bootstrap yields uncertainty only after training, so it cannot regularize learning. *Solution: online estimation* via the twins' parameter divergence, providing a per-step signal used during optimization.

## 3.3 Method Overview

We train two identical models, $M_1$ and $M_2$, initialized with the same parameters $w_1 = w_2$. At the start, we form two independent bootstrap datasets by sampling with replacement from the training set: $D_1^*$ and $D_2^*$. Training proceeds on paired mini-batches (or on paired full datasets for full-batch gradient descent): $(b_1 \in D_1^*, b_2 \in D_2^*)$. Parameters are partitioned into groups (e.g., layers) indexed by $\ell$ with sizes $D_\ell$. After each update, we compute a group-wise uncertainty from the twins' divergence,

$$\sigma_\ell^2 = \frac{1}{2D_\ell} \left\| w_{1,\ell} - w_{2,\ell} \right\|_2^2,$$

which acts as an online two-sample estimator of local parameter variance due to dataset resampling. During training, we use this estimate to sample weights per group,

$$\tilde{w}_\ell^{(i)} \sim \mathcal{N}\big(w_\ell^{(i)}, I\,\sigma_\ell^2\big), \quad i \in \{1, 2\},$$

so that the noise scale adapts from the resampling-induced uncertainty and provides a training-time estimator of bootstrap uncertainty that regularizes learning.

In complex optimization landscapes, independent models may drift to different minima. To confine exploration to a single solution basin, we periodically perform a mean-reset at scheduled intervals $K$: for each group,

$$w_{1,\ell}, w_{2,\ell} \overset{\text{i.i.d.}}{\sim} \mathcal{N}\Big(\tfrac{w_{1,\ell} + w_{2,\ell}}{2}, I\,\sigma_\ell^2\Big).$$

Independent sampling around the mean maintains i.i.d. trajectories while preventing inter-basin drift, so $\sigma_\ell^2$ reflects within-basin uncertainty rather than distances between distinct minima.

For inference, a simple deterministic option uses the mean of the twins' weights, $\frac{w_1+w_2}{2}$. When predictive uncertainty is required, one can perform Monte Carlo inference by sampling weights around the mean per group using $\sigma_\ell^2$ and averaging predictions.

---

**Algorithm 1** Twin-Bootstrap Gradient Descent

---

**Require:** Dataset $D$, epochs $E$, mini-batch size $B$, reset interval $K$; parameter groups $\{\ell\}$ with sizes $\{D_\ell\}$ (e.g., layers)
1: Initialize twin models $M_1, M_2$ with identical weights $w_1, w_2$
2: Create bootstrapped datasets $D_1^* \leftarrow \text{BOOTSTRAP}(D)$, $D_2^* \leftarrow \text{BOOTSTRAP}(D)$
3: Initialize group uncertainties $\sigma_\ell^2 \leftarrow 0$ for all $\ell$
4: **for** epoch $e = 1$ to $E$ **do**
5:     **for** each paired mini-batch $(b_1 \in D_1^*,\ b_2 \in D_2^*)$ **do**
6:         **Training-time sampling:** For each group $\ell$, sample $\varepsilon_\ell^{(1)}, \varepsilon_\ell^{(2)} \sim \mathcal{N}(0, I\,\sigma_\ell^2)$ and set $\tilde{w}_\ell^{(1)} \leftarrow w_{1,\ell} + \varepsilon_\ell^{(1)}$, $\tilde{w}_\ell^{(2)} \leftarrow w_{2,\ell} + \varepsilon_\ell^{(2)}$
7:         $L_1 \leftarrow L\Big(\{\tilde{w}_\ell^{(1)}\}_\ell; b_1\Big),\quad L_2 \leftarrow L\Big(\{\tilde{w}_\ell^{(2)}\}_\ell; b_2\Big)$
8:         $g_1 \leftarrow \nabla_{w_1} L_1; g_2 \leftarrow \nabla_{w_2} L_2$
9:         $w_1 \leftarrow \text{OPTIMIZER}(w_1, g_1); w_2 \leftarrow \text{OPTIMIZER}(w_2, g_2)$
10:         For each group $\ell$: $\sigma_\ell^2 \leftarrow \dfrac{1}{2D_\ell} \left\| w_{1,\ell} - w_{2,\ell} \right\|_2^2$
11:     **end for**
12:     **if** $e \bmod K = 0$ **then**
13:         For each group $\ell$: $w_{1,\ell}, w_{2,\ell} \overset{\text{i.i.d.}}{\sim} \mathcal{N}\left( \dfrac{w_{1,\ell} + w_{2,\ell}}{2},\ I\,\sigma_\ell^2 \right)$
14:     **end if**
15: **end for**

---

### 3.4 Theoretical Justification & Online Uncertainty

We justify the use of the twins' squared distance as an online estimate of local parameter uncertainty. Let $w^*$ denote the optimal parameters obtained by training on a bootstrap sample of the dataset $D$. Because the twins are trained on independent bootstrap samples and are periodically reset to remain in the same basin, their parameters $w_1$ and $w_2$ can be treated as i.i.d. draws from the (local) bootstrap distribution of $w^*$. Hence $\mathbb{E}[\|w_1 - w_2\|_2^2] = 2\,\text{Var}(w^*)$. For a parameter tensor with $D$ entries, this yields the per-parameter online estimator $\sigma_{\text{avg}}^2 = \frac{1}{2D} \|w_1 - w_2\|_2^2$, which provides a low-variance, online measure of local uncertainty and motivates the stochastic forward sampling used for regularization.

Viewed statistically, $\frac{1}{2D} \|w_1 - w_2\|_2^2$ is a *two-sample estimator* of the per-parameter variance $\text{Var}(w^*)$. While higher-variance than a many-sample bootstrap, it is unbiased, computable online, and—when aggregated over parameter groups (e.g., layers)—provides a stable training signal. The sampling-based mean-resets keep the twins in the same basin, making the i.i.d. assumption locally valid.

For grouped parameters (e.g., layers), we analogously obtain $\sigma_\ell^2 = \frac{1}{2D_\ell} \|w_{1,\ell} - w_{2,\ell}\|_2^2$, which is used for layer-wise forward sampling and resets in our neural network experiments. In addition to layer-wise grouping, we also evaluated a per-unit grouping (one group per neuron/channel) and observed virtually identical results.

The mean-reset mechanism ensures that this two-sample estimate remains valid. By periodically collapsing the twins, the procedure keeps them within the same solution basin, exploring its local geometry rather than diverging across different minima. Under this constraint, the measured parameter divergence reflects the uncertainty of the parameters within that specific basin, not the distance between distinct solutions.

**Estimator variance: two-sample vs. classical $B$-sample.** Assuming an approximately normal local bootstrap distribution and independence within a group, let the true per-parameter variance be $\tau^2 = \text{Var}(w^*)$. For a single parameter, the two-sample estimator $\hat{\sigma}^2 = \frac{1}{2}(w_1 - w_2)^2$ is unbiased with

$\mathrm{Var}(\hat{\sigma}^2) = 2\tau^4$; whereas the classical $B$-sample (sample-variance) estimator satisfies $\mathrm{Var}(\hat{\sigma}_B^2) = \frac{2\tau^4}{B-1}$. For a parameter group of size $D_\ell$ with isotropic variance $\tau_\ell^2$, averaging across entries reduces the two-sample estimator's variance by $1/D_\ell$: $\mathrm{Var}(\hat{\sigma}_\ell^2) = \frac{2\tau_\ell^4}{D_\ell}$. This grouped two-sample estimator is the quantity we use in practice. Grouping (e.g., per layer) yields a stable, low-variance online signal that scales to large models with about a $2\times$ cost.

We treat $w_1$ and $w_2$ as approximate i.i.d. draws from the local bootstrap distribution within a single solution basin. While this is a simplifying assumption about the training dynamics, our empirical results in complex landscapes suggest it is a practically effective model.

### 3.5 THE MEAN-RESET & STOCHASTIC REGULARIZATION

The implementation of the mean-reset mechanism itself required solving two practical issues. First, a simple average of the twins' weights could cause the models to overfit to the combined data from their separate bootstrap samples. To prevent this and preserve the statistical independence of their trajectories, we introduce model sampling at reset. Instead of setting both models to the same mean, the twin weights are reset independently to new values sampled from a distribution centered on their mean, with a standard deviation given by the current uncertainty estimate: resetting as $w_{1,\ell}, w_{2,\ell} \overset{\text{i.i.d.}}{\sim} \mathcal{N}(\frac{w_{1,\ell}+w_{2,\ell}}{2}, I\,\sigma_\ell^2)$ for each group $\ell$. This sampling-based reset maintains the expected distance distribution of the twins and prevents the unintended transfer of information. Second, in practice resets should be frequent early to keep both twins in the same basin, and can be spaced out later once trajectories stabilize.

The online uncertainty also serves as a training-time regularizer. At each training step, the weights of each twin model are stochastically sampled from a distribution centered on its current weight vector, with a standard deviation given by the online uncertainty estimate: At each step we sample $\tilde{w}_\ell^{(i)} \sim \mathcal{N}(w_\ell^{(i)}, I\,\sigma_\ell^2)$ for $i \in \{1, 2\}$. This stochasticity forces the model to be robust to variations in its parameters, naturally encouraging it to learn flatter, more generalizable minima (Hochreiter & Schmidhuber, 1997; Keskar et al., 2017; Foret et al., 2021; Izmailov et al., 2018). Conceptually, this resembles training with weight noise (Bishop, 1995) and techniques used for predictive uncertainty (e.g., Bayesian neural networks (Blundell et al., 2015)), but here the noise scale is estimated online from two resampled trajectories.

### 3.6 INFERENCE (TEST-TIME) PROCEDURE

After training, a simple option is a deterministic inference procedure: use the mean of the twins' weights and do not sample. Given an input $x$, the deterministic prediction is $\hat{y} = f(x; \frac{w_1+w_2}{2})$.

For applications that require predictive uncertainty, one can perform Monte Carlo (MC) inference by sampling weights around the mean and averaging predictions (Kendall & Gal, 2017; Guo et al., 2017): we draw $w^{(s)} \sim \mathcal{N}(\frac{w_1+w_2}{2}, I\,\sigma_{\text{avg}}^2)$ and average predictions $\hat{y}_{\text{MC}} = \frac{1}{S}\sum_{s=1}^{S} f(x; w^{(s)})$, with an optional uncertainty estimate from the sample variance of $\{f(x; w^{(s)})\}_{s=1}^{S}$. For classification, average class probabilities across samples.

## 4 EMPIRICAL VALIDATION

The empirical validation is structured to first demonstrate that our computationally efficient "twin" model approach yields a valid online uncertainty estimate. We then show how our method overcomes the primary theoretical barrier to its application in complex landscapes, and finally, we showcase its advantages and utility on high-dimensional problems. Our goal here is a method-and-theory contribution that links classical resampling to modern deep networks; results are therefore representative rather than the outcome of extensive performance tuning.

### 4.1 VALIDATING MECHANISMS ON TOY LANDSCAPES

The proposal of this work is to transform bootstrapping from a post-hoc analysis into an integral component of the optimization process. Classical bootstrapping is often dismissed for large-scale

problems due to the prohibitive cost of training hundreds of independent models to form a sampling distribution. We posit that a meaningful, low-variance estimate of uncertainty can be derived *online* from just two "twin" models.

To validate this assumption of computational efficiency, we first demonstrate the mechanism in a simple convex landscape with a clear ground truth, where two models estimate the mean of a 2D Gaussian data cloud. Figure 1 shows that the twin trajectories converge toward the true parameter center, while the online uncertainty estimate (circles) provides a live, per-step measure of local parameter variability that accurately tracks the true uncertainty of the estimator. This provides a visual proof that even with only two samples, we achieve online bootstrapping without the prohibitive cost of traditional methods.

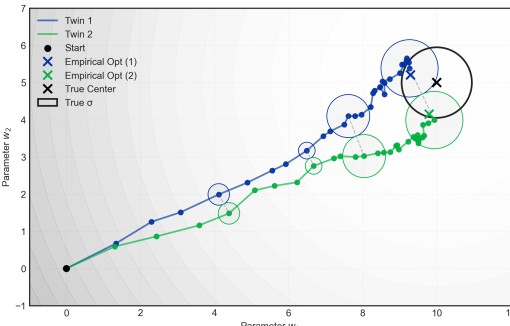

Figure 1: Two-dimensional Gaussian example with bootstrapped twin trajectories. Per-stride uncertainty circles capture local variability around the mean path; markers indicate empirical optima, true center, and true $\sigma$.

### 4.1.1 THE TWO-BASIN LANDSCAPE: OVERCOMING MULTI-MODALITY

The immediate difficulty for the online bootstrapping approach is its application to non-convex landscapes. In such settings, independent models are expected to diverge to different local minima, rendering their parameter variance a meaningless measure of inter-basin distance. We construct a landscape with two distinct minima to demonstrate how Twin-Boot's mean-reset mechanism is designed specifically to solve this fundamental problem. Without a reset, twin models diverge to separate minima; with the periodic, sampling-based mean-reset, the twins are constrained to a single basin and their divergence becomes a stable, informative measure of local uncertainty.

As illustrated in Figure 2, the reset acts as a constraint that co-locates the twins within a single basin while preserving i.i.d. trajectories via sampling around the mean. This confinement makes online bootstrapping viable in complex landscapes, ensuring that the measured divergence reflects the geometry of a single solution rather than the distance between distinct minima.

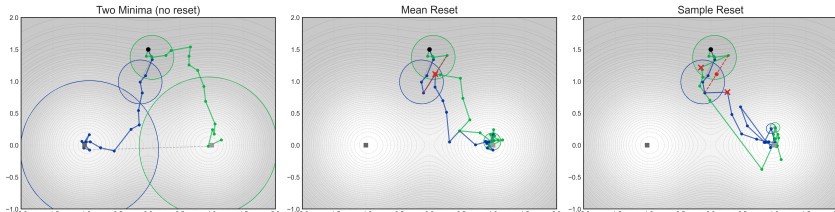

Figure 2: Two-basin triptych. Left: without mean-reset, twins trained on independent bootstraps diverge to separate minima, making $\sigma$ invalid as a local measure. Middle: periodic mean-reset to the mean keeps twins in a single basin. Right: sampling-based mean-reset maintains i.i.d. twin trajectories while confining exploration to one basin; the resulting divergence yields a stable, local uncertainty estimate.

### 4.1.2 Two-Basin Parameter Sweeps

We conduct sweeps on the two-basin landscape to probe robustness. Varying dataset size, data noise, learning rate, and mini-batch size, we summarize the final $\sigma$ over seeds (mean-reset only) and overlay a curvature-corrected theory (dashed). The estimate scales correctly with data variability ($\propto \sigma_{\text{data}}/\sqrt{M}$) while remaining largely invariant to optimizer settings, consistent with a geometry-aware uncertainty; see Appendix Fig. 5 for the complete visualization.

**Resampling-based uncertainty.** Because the uncertainty is driven by data resampling, not optimizer dynamics, the estimate is largely invariant to optimizer settings like learning rate and batch size. The estimate matches a curvature-corrected single-well reference derived from the local Hessian of one basin via the factor $S(\text{bw})$, which rescales the baseline $\sigma_{\text{data}}/\sqrt{M}$ by the principal curvatures $(\lambda_\perp, \lambda_\parallel)$. This explains both the invariance to optimizer settings and the correct scaling with data size and noise, without computing Hessians explicitly.

## 4.2 Application to Deep Networks

While classical bootstrapping is computationally infeasible for deep neural networks, the multimodal nature of their loss landscapes presents a greater theoretical barrier. The mean-reset mechanism overcomes this, enabling the use of online bootstrapping to regularize and improve the calibration of deep networks on CIFAR-10 using the full 50k training images. The online uncertainty estimate is used to inject noise during training, acting as an adaptive regularizer.

We employ a *grouped two-sample estimator* of parameter variance (e.g., per layer), whose variance scales as $O(1/D_\ell)$. This provides a stable online uncertainty signal with about a $2\times$ training cost, making the method tractable for large networks.

**Training protocol for neural networks.** We follow the Twin-Boot procedure end-to-end as instantiated for neural networks:

1. Initialize two identical models with the same weights, $w_1 = w_2$.

2. At the start of training, form two independent bootstrap datasets by sampling *with replacement* from the 50k-image CIFAR-10 training set: $D_1^*, D_2^*$.

3. Iterate through paired mini-batches ($b_1 \in D_1^*$, $b_2 \in D_2^*$). For each forward pass of each twin, sample weights *per layer* from an isotropic Gaussian centered at the current weights with a layer-specific scale $\sigma_\ell$:

$$\tilde{w}_\ell^{(i)} \sim \mathcal{N}\left(w_\ell^{(i)}, I\,\sigma_\ell^2\right), \quad i \in \{1, 2\}.$$

4. Compute losses on the paired batches and update each twin with a standard optimizer.

5. Update the online, layer-wise uncertainty from the twins' parameter divergence:

$$\sigma_\ell^2 \leftarrow \frac{1}{2D_\ell}\left\|w_{1,\ell} - w_{2,\ell}\right\|_2^2,$$

where $D_\ell$ is the number of parameters in layer $\ell$.

6. At scheduled intervals $K$, perform a sampling-based mean-reset: for each layer

$$w_{1,\ell},\, w_{2,\ell} \overset{\text{i.i.d.}}{\sim} \mathcal{N}\left(\tfrac{w_{1,\ell}+w_{2,\ell}}{2}, I\,\sigma_\ell^2\right).$$

The bootstrap datasets $D_1^*, D_2^*$ are not modified by resets.

Figure 3 summarizes performance and calibration on CIFAR-10. The left panel shows training/validation accuracy for the baseline (blue) and Twin-Boot (green); Twin-Boot narrows the generalization gap and attains higher validation accuracy. The right panel is a reliability diagram comparing baseline, a 2-member ensemble, and Twin-Boot; Twin-Boot substantially reduces miscalibration relative to the baseline and approaches the ensemble.

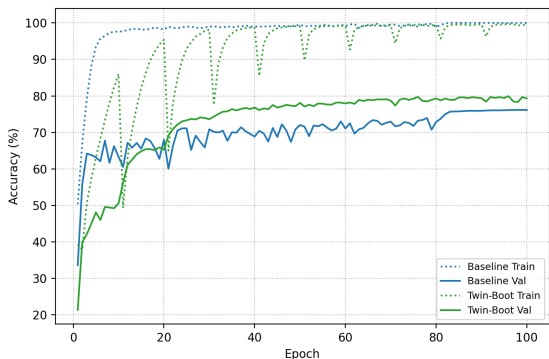 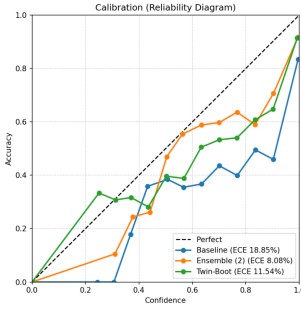

Figure 3: CIFAR-10 results. Left: training and validation accuracy for the baseline (blue) and Twin-Boot (green). Right: calibration (reliability diagram) comparing baseline, a 2-member ensemble, and Twin-Boot.

**CIFAR-10 results summary.** We used an early-heavy reset schedule: resets at epochs 1 and 2, then every 10 epochs thereafter. This keeps the twins in the same local solution neighborhood during the volatile early phase, and requires less frequent synchronization once trajectories stabilize. Table 1 reports accuracy, calibration (ECE), and relative cost. Twin-Boot attains higher accuracy than the single-model baseline and markedly better calibration. At comparable cost it also achieves higher validation accuracy than a 2-member ensemble, while acting *during* training rather than post hoc—the online uncertainty signal regularizes learning step by step; its calibration approaches the ensemble. Our focus is the method and its statistical link to classic bootstrapping; we did not tune aggressively for peak performance.

Table 1: CIFAR-10 Performance Comparison.

| Method | Test Acc. (%) | ECE (%) | Relative Cost |
|---|---|---|---|
| Baseline (1 model) | 76.11 | 18.73 | $1.0\times$ |
| 2-Member Deep Ensemble | 78.84 | **8.08** | $2.0\times$ |
| **Twin-Boot (ours)** | **79.46** | 11.34 | $\approx 2.0\times$ |
| Ablation: No Sampling | 77.51 | 14.81 | $\approx 2.0\times$ |

The online uncertainty exhibits a characteristic pattern: a brief high-variance phase early in training, followed by convergence to a consistent layer-wise structure (highest in later layers). A full visualization is provided in Appendix Fig. 6.

### 4.3 APPLICATION TO A SCIENTIFIC INVERSE PROBLEM

To evaluate generality beyond standard benchmarks, we consider a nonlinear seismic inversion task: infer a 2D subsurface velocity map $v \in \mathbb{R}^P$ from sparse measurements $y \in \mathbb{R}^M$ produced by a known forward operator (Appendix A.6). In our setup, $P = 900$ (a $30 \times 30$ grid) and $M = 4096$. Each measurement corresponds to a normalized 2D Gaussian kernel centered at a random location applied to the field, followed by an element-wise nonlinearity $f(v) = \tanh(\beta v)$ and additive noise. The problem is ill-posed and multi-modal; overfitting is common with standard optimizers. Using the proposed training procedure, we obtain lower test loss and reconstruction error with modest overhead ($1.4\times$).

The online uncertainty estimate yields a spatial map. The learned $\sigma$ highlights areas where the reconstruction is least reliable due to limited information, aligning with absolute error maps (Figure 4). In these simulations, parameters were grouped into $3 \times 3$ patches (one $\sigma$ per patch) to obtain more precise local uncertainty estimates.

Table 2: Twin-Boot vs. Standard Optimizer on Nonlinear Seismic Inversion (mean $\pm$ 95% CI over 25 seeds).

| Mode | Train Loss | Test Loss | Recon MSE | Time/run (s) |
|------|-----------|-----------|-----------|--------------|
| Twin-Boot | $0.0007 \pm 0.0004$ | $0.0032 \pm 0.0011$ | $0.0098 \pm 0.0014$ | $8.47 \pm 0.11$ |
| Standard | $0.0002 \pm 0.0000$ | $0.0315 \pm 0.0150$ | $0.0338 \pm 0.0058$ | $6.01 \pm 0.08$ |

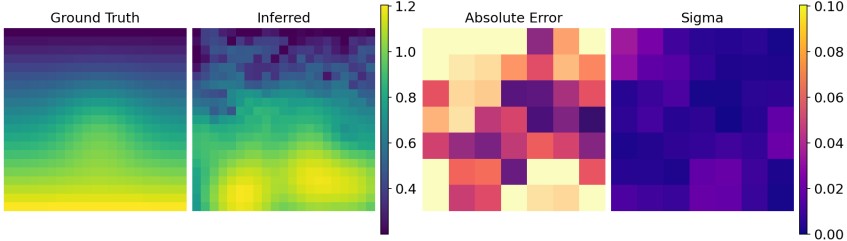

Figure 4: Seismic inversion: ground truth, inferred, absolute error, and $\sigma$ maps. Uncertainty concentrates on poorly constrained regions and correlates with reconstruction error.

## 5 CONCLUSION

We introduced Twin-Bootstrap Gradient Descent, which integrates resampling-based uncertainty into gradient descent. The approach combines three components—independently bootstrapped training trajectories, a two-model design, and periodic mean-resets that maintain within-basin exploration. Together these elements allow the twins' divergence to serve as an online estimate of local parameter uncertainty and use it to guide training.

Our empirical validation spans low-dimensional landscapes, a standard deep-learning benchmark, and a seismic inverse problem. The experiments confirm basin confinement under mean-resets, show that the uncertainty signal reflects local geometry, and demonstrate that uncertainty-driven stochasticity acts as an effective regularizer. In settings that benefit from interpretable or mechanistic models, the learned uncertainty map provides a meaningful companion to the primary reconstruction.

This work shows that principles from statistical resampling can be effectively integrated into modern optimization frameworks. By reformulating bootstrapping as an online two-sample estimator, finite-sample variability becomes a direct signal for regularizing complex models. This links the statistical properties of the data and the geometric properties of the solutions found by the optimizer, providing a general framework for uncertainty-aware optimization.

**Limitations.** The two-sample estimator has higher variance for very small parameter groups; aggregating at the layer level reduces variance, but this limitation remains for fine-grained groupings. Basin confinement can fail if resets are too infrequent or too weak, which may bias the uncertainty estimate. The method is sensitive to the reset schedule and the cadence of weight sampling. Finally, training incurs roughly a $2\times$ compute and memory overhead due to maintaining two models and performing training-time sampling.

**Future Work.** Twin-Boot opens several promising avenues for future research. A deeper theoretical analysis of the mean-reset mechanism could provide a more formal guarantee of its basin-confining properties. Exploring its application to domains where calibration and interpretability are central—such as structured prediction, medical imaging, and scientific machine learning—could uncover new benefits. Finally, investigating the relationship between the two-sample uncertainty estimate used here and Bayesian posterior uncertainty could clarify when the measures agree, when they differ, and how they might be combined.

## REPRODUCIBILITY STATEMENT

We provide the full training procedure in Algorithm 1 and detail assumptions and mechanisms in Sections 3.4–3.5. Experimental setups and hyperparameters for CIFAR-10 and the seismic task are described in Section 4.2, with additional implementation details and ablations in the Appendix ("Implementation Details" and "Additional CIFAR-10 Ablations"). A code repository with training scripts, configuration files, and figure-generation utilities will be published upon acceptance.

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

## A  IMPLEMENTATION DETAILS

This appendix provides details to reproduce our empirical results.
**Hardware:** All experiments were performed on a single NVIDIA T4 GPU.

### A.1 TOY LANDSCAPES

**Loss functions:** Analytically defined 2D landscapes (Gaussian and two-basins) to enable precise visualization of trajectories and local geometry.

**Setup:** Two identical models (twins) are trained on independent bootstrap samples drawn once at the start of training. We visualize trajectories, per-stride uncertainty circles derived from the online estimate, and reference markers (true center and variance for Gaussian; basin minima and ridge structure for two-basins).

**Mean-reset mechanism:** We compare three modes on the two-basin landscape: (i) no reset, where twins drift to different minima and $\sigma$ ceases to reflect local uncertainty; (ii) deterministic mean reset that co-locates twins; and (iii) sampling-based mean reset that preserves i.i.d. trajectories while confining exploration to a single basin, yielding a stable local uncertainty estimate.

**Parameter sweeps:** On the two-basin landscape, we vary dataset size, data noise, learning rate, and mini-batch size (mean-reset only) and summarize the final uncertainty as mean $\pm$ standard deviation across seeds. A curvature-corrected single-well theory is used as a reference (details below). The estimate scales with data variability and remains robust across optimizer settings.

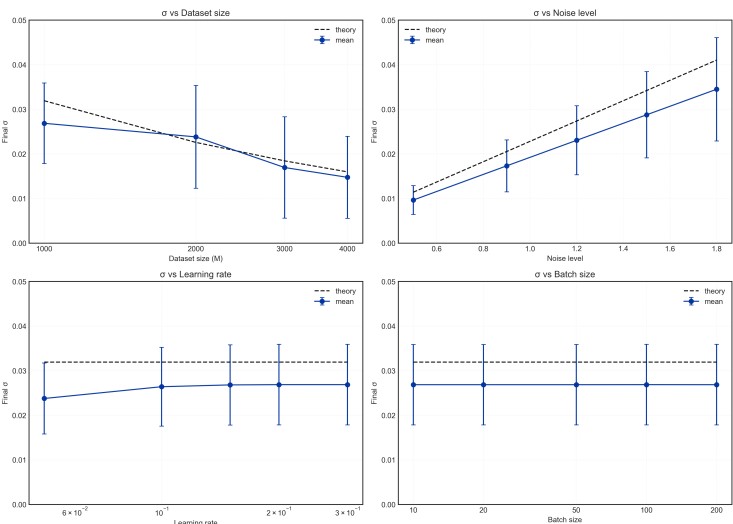

Figure 5: Two-basin sweeps (older 2x2 layout). Final $\sigma$ versus key parameters (dataset size, noise level, learning rate, batch size), reported as mean $\pm$ standard deviation across seeds. Dashed: curvature-corrected single-well theory $\sigma_{\text{theory}} = S(\text{bw})\,\sigma_{\text{data}}/\sqrt{M}$.

### A.2 GAUSSIAN LANDSCAPE

**Model:** A simple linear model with two parameters to find the mean of a 2D Gaussian data cloud.

**Optimizer:** SGD.

**Figure 1 settings (single run):** $N = 400$, variance per dimension $= 120$, epochs $= 5$, batch size $= 200$, learning rate $= 0.07$, uncertainty stride $= 4$, seed $= 7$, bootstrap ON with twin-specific bootstrap datasets.

**Theory reference:** For estimating a Gaussian mean, the per-parameter variance of the optimal estimator is $\text{Var}(w^*) = \sigma_{\text{data}}^2/M$, so the reference line is $\sigma_{\text{true}}(M) = \sigma_{\text{data}}/\sqrt{M}$. The twin-based online estimate targets this via $\sigma_{\text{avg}}^2 = \frac{1}{2D}\|w_1 - w_2\|_2^2$.

### A.3 TWO-BASIN LANDSCAPE (FIG. 2 AND APPENDIX FIG. 5)

**Landscape:** We use a symmetric two-well potential with minima at $\mu_1$ and $\mu_2$ separated by distance $d = 2$ and well width (standard deviation) $\sigma$:

$$L(w) \;=\; -\exp\left(-\frac{\|w-\mu_1\|^2}{2\sigma^2}\right) \;-\; \exp\left(-\frac{\|w-\mu_2\|^2}{2\sigma^2}\right).$$

**Curvature-corrected single-well scaling:** Modeling each basin locally by its Hessian yields the reference

$$\sigma_{\text{theory}}(M, \sigma_{\text{data}}; \text{bw}) \;=\; S(\text{bw})\,\frac{\sigma_{\text{data}}}{\sqrt{M}},$$

with $\varepsilon = \exp\left(-d^2/(2\sigma^2)\right)$, principal curvatures

$$\lambda_\perp = \tfrac{1+\varepsilon}{\sigma^2}, \qquad \lambda_\parallel = \tfrac{1+\varepsilon}{\sigma^2} - \varepsilon\,\tfrac{d^2}{\sigma^4}, \qquad S(\text{bw}) = \sqrt{\tfrac{1}{2}\left(\tfrac{1}{\lambda_\perp} + \tfrac{1}{\lambda_\parallel}\right)}.$$

This is the dashed reference in Appendix Fig. 5.

**Sweeps protocol (Appendix Fig. 5):** Mean-reset mode; summarize final $\sigma$ as mean $\pm$ standard deviation across seeds. Reset schedule: epochs $\{1, 2, 6, 12\}$. We varied dataset size, data noise, learning rate, and mini-batch size, trained for 40 epochs with a step learning-rate schedule.

## A.4 CIFAR-10

**Model:** VGG-16 backbone with weight-normalized convolutional and linear layers; uncertainty buffers (one $\sigma$ per output unit or per layer) drive training-time weight sampling; final classifier is a standard linear layer.
**Dataset:** Full CIFAR-10 training set with 50,000 images.
**Optimizer:** Adam.
**Hyperparameters:** Learning rate 0.001. Mini-batch size: 64. Reset schedule: epochs $\{1, 2\}$ and then every 10 epochs. We used layer-wise grouping for $\sigma_\ell$ and also tested unit-wise grouping with virtually identical results.

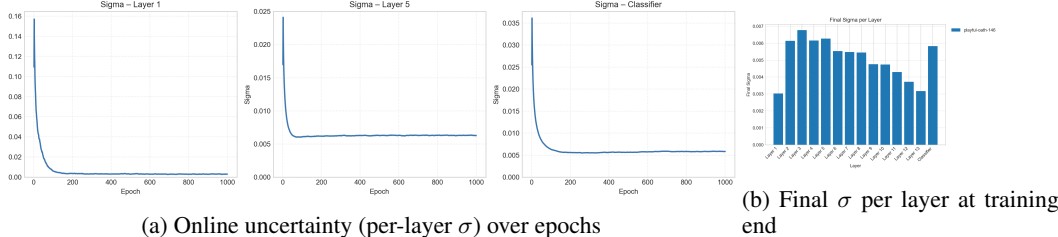

(a) Online uncertainty (per-layer $\sigma$) over epochs

(b) Final $\sigma$ per layer at training end

Figure 6: Neural network uncertainty stability and structure. Left: time series of layerwise $\sigma$ showing rapid early decay and a stable layer structure. Right: final $\sigma$ per layer for the same run.

## A.5 NONLINEAR SEISMIC INVERSION

**Model:** Parameter vector $v \in \mathbb{R}^P$ on a $30 \times 30$ grid ($P = 900$). The forward model is $y = Kf(v) + \varepsilon$, with $K \in \mathbb{R}^{4096 \times 900}$ whose rows are L2-normalized 2D Gaussian kernels at random centers, and $f$ either $\tanh(\beta v)$ (default) or a cubic proxy. Training uses full-batch gradient descent. Uncertainty $\sigma$ is estimated on $3 \times 3$ patches (one $\sigma$ per patch) and broadcast per block.
**Optimizer:** Adam.
**Hyperparameters:** Learning rate: started at 0.001 and decayed exponentially. Mini-batch size: 32. Reset interval ($K$): adaptive, with initial $K_0 = 50$.

**Benchmark details.** The seismic inversion task is a synthetic inverse problem designed to capture essential challenges of real-world geophysical imaging, such as Full-Waveform Inversion (FWI) (Tarantola, 1984), in a computationally tractable 2D setting. The goal is to reconstruct a 2D subsurface velocity map represented by a grid of parameters of size $P = 900$.

**Forward model.** The observations are generated according to

$$y = Kf(v) + \varepsilon,$$

where $v \in \mathbb{R}^P$ is the unknown velocity field, $y \in \mathbb{R}^M$ are the measurements, $K \in \mathbb{R}^{M \times P}$ is a linear measurement operator, and $f(\cdot)$ is an element-wise nonlinear function. The operator $K$ consists of $M = 4096$ rows, each a normalized 2D Gaussian kernel centered at a random location, simulating sparse measurements of the field. The nonlinearity $f(v) = \tanh(\beta v)$ models complex wave responses, and $\varepsilon$ denotes measurement noise.

**Problem characteristics.** The problem is nonlinear due to $f(\cdot)$; ill-posed and multi-modal because multiple distinct $v$ can explain $y$; and effectively over-parameterized—despite $M > P$—because of the nonlinearity and ill-posedness, which make overfitting a central concern for standard optimizers.

## B  LLM Usage

We used large language model (LLM) tools to (i) assist writing and revision (drafting, restructuring, and clarity edits), (ii) support coding (implementation guidance, refactoring, and debugging with standard coding assistants), and (iii) provide feedback and discussion on design choices and presentation.

