# OpenReview forum: "Uncertainty-Aware Gradient Descent via Online Bootstrapping"
_ICLR.cc/2026/Conference — ICLR 2026 Conference Withdrawn Submission_

### Official Review · Reviewer_Eo6i · 2025-10-27

**Soundness:** 3
**Presentation:** 4
**Contribution:** 2
**Rating:** 4
**Confidence:** 3

**Summary:**

The paper proposes Twin-Bootstrap Gradient Descent, a way to integrate classical bootstrapping into optimization. Tested on datasets like cifar-10

**Strengths:**

- Originality: Novel integration of bootstrapping into optimization loop (online vs post-hoc)

- Quality: Thoughtful experiments to validate the different mechanisms

- Clarity: Well-motivated problem, clearly written paper

- Significance: Addresses important calibration problem & more efficient than full bootstrapping

**Weaknesses:**

- Insufficient experiments: only CIFAR-10/VGG-16, which are very small scale for modern CV. At minimum would need imagenet scale and bigger models, at least ResNets or vision transformers.

- Missing Baselines: missing MC Dropout, SWAG, variational inference, temperature scaling,

- CIFAR-10 underperformance: 79.46% seems very low. VGG models can get 90+ scores, so does this harm performance

- Limited ablations: more ablations would be nice to better understand the mechanisms

- Insights: No characterization when Twin-Boot helps vs fails - why good on seismic but modest on CIFAR-10?

**Questions:**

- Larger scale: Have you tested on ImageNet size datasets and bigger models ResNets, Transformers? Is there a specific reason why this can’t scale

- Baseline comparisons: please can you add the aforementioned baselines

- Low performance: why is cifar-10 so low?

- When does it help: Can you characterize problem properties where Twin-Boot excels? Like where does it do well and where is it weak?

---

### Official Review · Reviewer_u8E2 · 2025-11-03

**Soundness:** 1
**Presentation:** 2
**Contribution:** 1
**Rating:** 2
**Confidence:** 4

**Summary:**

The paper proposes Twin-Boot, a dual-network optimization scheme where two models are trained in parallel on independent mini-batches; their divergence approximates bootstrap-style uncertainty, and periodic mean-resets keep them in the same basin. This uncertainty is then used to inject adaptive noise into gradient descent. Experiments on toy tasks, CIFAR-10, and a seismic inversion example show moderate gains.

**Strengths:**

Simple, clearly described algorithm combining ensemble variance with uncertainty-aware regularization.

Intuitive visualization of uncertainty evolution; clean toy experiments demonstrate behavior.

Good connection to the bootstrap idea and to uncertainty calibration literature.

**Weaknesses:**

1. Conceptual confusion around “uncertainty.”

The central framing—estimating uncertainty during training—is ill-defined.
The difference between two evolving parameter sets does not represent epistemic or predictive uncertainty in the Bayesian or ensemble sense; it merely reflects optimization instability.
Bootstrapping assumes independent fits on fixed data; here both models share the same data distribution and correlated stochastic gradients.
Thus, the quantity being measured is better interpreted as gradient variance or local training noise, not genuine uncertainty.
This makes the “uncertainty-aware” label misleading—the method is a form of noise-scaled regularization, not uncertainty quantification.

2. Very similar to existing two-network and teacher-student frameworks.

Twin-Boot’s structure closely mirrors well-known methods designed for learning under noisy or unstable supervision, where two models are kept partially synchronized to stabilize updates:

Mean Teacher (Tarvainen & Valpola, 2017) — maintains a student and an EMA teacher; both are kept close to filter noise in targets.

Co-teaching / DivideMix (Han et al., 2018; Li et al., 2020) — two networks train each other on data they believe to be clean, relying on disagreement as a signal of uncertainty.

Bootstrapped DQN (Osband et al., 2016) — parallel learners estimate epistemic uncertainty via disagreement.

SWA / SAM (Izmailov et al., 2018; Foret et al., 2021) — use weight averaging or perturbation to smooth optimization.
The mean-reset in Twin-Boot plays almost the same stabilizing role as the EMA in Mean Teacher, except the paper reinterprets this stabilization term as “bootstrap consistency.” The resemblance to these prior approaches is so strong that the claimed novelty—“online bootstrapping for uncertainty-aware optimization”—is largely a change in terminology, not mechanism.

3. Theoretical claims are heuristic.

Treating the twin divergence as a bootstrap variance has no formal grounding; both networks are correlated and do not correspond to independent resamples. No analysis links this divergence to predictive calibration or Bayesian posterior spread.

4. Experimental validation is limited.

Experiments are small-scale and exclude the most relevant baselines (SWAG, SGLD, SAM, or teacher-student noise-robust training). Reported gains (~0.5–1%) do not justify doubling computation, and no calibration or robustness metrics are included despite the uncertainty claim.

**Questions:**

Meaning of “uncertainty.”
What exactly does the twin divergence represent? Is it epistemic, gradient variance, or just optimization noise? Without a clear probabilistic interpretation, the term uncertainty feels misplaced.

Bootstrap justification.
Since both models share the same data and correlated gradients, how is this a valid bootstrap estimator? Show evidence that twin variance correlates with predictive uncertainty or calibration metrics.

Relation to teacher–student frameworks.
The setup and mean-reset are strikingly similar to Mean Teacher and co-training methods for noisy or semi-supervised learning. How is this different beyond terminology?

Ablation and mechanism.
What part of the method actually improves performance—the dual setup, mean-reset, or noise scaling? Provide controlled comparisons.

Missing baselines.
Why exclude direct competitors like SWAG, SGLD, or SAM, which already handle noise or uncertainty during optimization?

---

### Official Review · Reviewer_EHLs · 2025-11-03

**Soundness:** 3
**Presentation:** 2
**Contribution:** 2
**Rating:** 2
**Confidence:** 4

**Summary:**

This manuscript addresses the limitation of standard SGD, which only provides point estimates of model parameters without corresponding uncertainty quantification. Borrowing ideas from bootstrapping, the authors introduce a twin-bootstrap strategy to integrate uncertainty estimation into the optimization process through resampling. This method successfully identifies more robust and flatter solutions. The effectiveness of the method is demonstrated through experiments on both toy datasets and the CIFAR-10 benchmark.

**Strengths:**

This manuscript presents a novel model learning and optimization framework capable of performing parameter uncertainty estimation and utilizing this information to guide the optimization process. A key theoretical innovation lies in its reformulation of classical bootstrapping as an online two-sample estimator. This conceptual shift effectively turns what is typically a post-hoc analysis into an actionable signal that directly informs the optimization trajectory.

**Weaknesses:**

The manuscript currently contains several issues regarding the formulation of equations and definition of symbols that require refinement. Key experimental settings are not clearly specified, which hinders the reproducibility of the reported results. Furthermore, the theoretically claimed capabilities—such as favoring flatter solutions and improving generalization—are not sufficiently validated in the experimental section. More rigorous ablation studies and comparative analyses are needed to substantiate these claims.

**Questions:**

1.In Section 3.1, the letter 'l' is used to denote multiple distinct variables, leading to notational ambiguity.

2.The equations in the manuscript are currently unnumbered, which hinders cross-referencing and discussion. Furthermore, the meaning of the superscript in Line 152 appears inconsistent with the subscript notation used in the equation on Line 148, creating ambiguity.

3.Line 206, it is stated that per-unit grouping yields identical results to other methods. However, it would be valuable to discuss its potential disadvantages compared to alternatives like layer-wise grouping. The authors are also encouraged to offer practical advice for future researchers on selecting grouping strategies.

4.The selection of the hyperparameter K appears empirical. We recommend conducting an ablation study to systematically evaluate its impact on final performance.

5.What kind of noise are investigated in section 4.1.2?

6.For the experiments in Section 4.2, the sizes of the subsets D*_1 and D*_2 should be specified. Please provide the rationale for the chosen sizes and clarify whether any pre-trained models were utilized in this section.

7.The manuscript does not sufficiently demonstrate the advantages of the proposed method over a naive ensemble of two independent models. A comparative analysis is needed to justify the choice of the proposed framework over this simpler baseline.

8.The theoretical potential of the proposed method to escape local minima and converge towards a global optimum should be discussed, with supporting evidence from the empirical results or relevant literature.

9.The claimed generalization ability is not conclusively demonstrated by the experimental results. More rigorous evidence, such as testing on more diverse or challenging benchmarks, is suggested to substantiate this claim.

---

### Author Response · Authors · 2025-12-02
**Withdrawal by Authors**

Dear Area Chair and Reviewers,

We appreciate the time you took to review our work and your detailed feedback. We are encouraged by the positive reception of the core theory, method, and novel idea of the paper. Reviewers specifically recognized the novelty of integrating bootstrapping directly into the optimization loop, found the algorithm simple and clear, and appreciated the intuitive visualization of the uncertainty evolution.

We have decided to withdraw the submission to perform a revision of our experimental suite. Based on the feedback, we will focus on broader comparisons, clarifying the contribution of the different noise sources, and more illustrative examples where explicit parameter-space (epistemic) uncertainty is critical.

Thank you again for your service to the community.

Sincerely,

The Authors

---

### Note · Authors · 2025-12-02

I have read and agree with the venue's withdrawal policy on behalf of myself and my co-authors.